# Efficacy Validation of SARS-CoV-2-Inactivation and Viral Genome Stability in Saliva by a Guanidine Hydrochloride and Surfactant-Based Virus Lysis/Transport Buffer

**DOI:** 10.3390/v15020509

**Published:** 2023-02-11

**Authors:** James Gitau Komu, Dulamjav Jamsransuren, Sachiko Matsuda, Haruko Ogawa, Yohei Takeda

**Affiliations:** 1Graduate School of Animal and Veterinary Sciences and Agriculture, Obihiro University of Agriculture and Veterinary Medicine, 2-11 Inada, Obihiro 080-8555, Hokkaido, Japan; 2Department of Medical Laboratory Sciences, College of Health Sciences, Jomo Kenyatta University of Agriculture and Technology, Nairobi P.O. Box 62000-00200, Kenya; 3Department of Veterinary Medicine, Obihiro University of Agriculture and Veterinary Medicine, 2-11 Inada, Obihiro 080-8555, Hokkaido, Japan; 4Research Center for Global Agromedicine, Obihiro University of Agriculture and Veterinary Medicine, 2-11 Inada, Obihiro 080-8555, Hokkaido, Japan

**Keywords:** SARS-CoV-2-inactivation, virus lysis/transport buffer, RNA stability, saliva, guanidine hydrochloride

## Abstract

To enhance biosafety and reliability in SARS-CoV-2 molecular diagnosis, virus lysis/transport buffers should inactivate the virus and preserve viral RNA under various conditions. Herein, we evaluated the SARS-CoV-2-inactivating activity of guanidine hydrochloride (GuHCl)- and surfactant (hexadecyltrimethylammonium chloride (Hexa-DTMC))-based buffer, Prep Buffer A, (Precision System Science Co., Ltd., Matsudo, Japan) and its efficacy in maintaining the stability of viral RNA at different temperatures using the traditional real-time one-step RT-PCR and geneLEAD VIII sample-to-result platform. Although Prep Buffer A successfully inactivated SARS-CoV-2 in solutions with high and low organic substance loading, there was considerable viral genome degradation at 35 °C compared with that at 4 °C. The individual roles of GuHCl and Hexa-DTMC in virus inactivation and virus genome stability at 35 °C were clarified. Hexa-DTMC alone (0.384%), but not 1.5 M GuHCl alone, exhibited considerable virucidal activity, suggesting that it was essential for potently inactivating SARS-CoV-2 using Prep Buffer A. GuHCl and Hexa-DTMC individually reduced the viral copy numbers to the same degree as Prep Buffer A. Although both components inhibited RNase activity, Hexa-DTMC, but not GuHCl, directly destroyed naked viral RNA. Our findings suggest that samples collected in Prep Buffer A should be stored at 4 °C when RT-PCR will not be performed for several days.

## 1. Introduction

Since its discovery in Wuhan, China in December 2019, severe acute respiratory syndrome coronavirus 2 (SARS-CoV-2) has rapidly resulted in one of the deadliest pandemics, constituting a global health emergency [1,2]. SARS-CoV-2, which belongs to the genus *Betacoronavirus*, is an enveloped, positive-sense, single-stranded RNA virus [3]. According to the WHO Coronavirus (COVID-19) Dashboard, as of January 2023, more than 752 million infections and a mortality rate of 0.9% have been reported worldwide [4].

The diagnosis of SARS-CoV-2 infection involves the detection of nucleic acids, antibodies, or viral proteins, as reviewed elsewhere [5]. Real-time reverse transcription polymerase chain reaction (RT-PCR) using respiratory samples has been the gold standard for SARS-CoV-2 diagnosis [6,7]. Real-time RT-PCR has attracted substantial attention for the diagnosis of SARS-CoV-2 infection because it permits rapid and high-throughput screening in patients, which is critical during a public health emergency [8]. Although respiratory tract samples were initially recommended for the diagnosis of SARS-CoV-2 infection using real-time RT-PCR [9], saliva is currently accepted as an alternative sample with good sensitivity and specificity in the detection of both viral RNA and antigen [10,11,12]. Indeed, there are reports of higher levels and prolonged detection of viral RNA in saliva compared to nasopharyngeal swabs in paired sample studies [13,14].

In saliva, SARS-CoV-2 RNA is relatively stable for long periods at 4 °C, for more than 7 days at approximately 16–19 °C, and for more than 3 days at 30 °C [15,16]. In particular, SARS-CoV-2 RNA-positive samples with lower cycle threshold (Ct) values showed viral RNA stability after prolonged storage [16]. Because this report suggested that infectivity is also maintained for prolonged periods, strict biosafety procedures are required during sample collection, transportation, and processing for diagnosing SARS-CoV-2 infection, which is critical for preventing accidental exposures [17,18]. Several approaches have been tested in vitro for inactivating SARS-CoV-2 including electromagnetic radiation [19,20,21,22], heat [19,23,24], and chemicals such as alcohols, hydrogen peroxide, surfactants, aldehydes, and guanidine-based buffers, which are used as disinfectants, fixatives, and lysis buffers [19,25,26,27].

At present, many commercially available virus lysis and transport buffers (virus lysis/transport buffer) with inactivation potential against SARS-CoV-2 in various samples are available. However, the composition and/or concentrations of active compounds remain undisclosed for most buffers even though they are composed of readily available laboratory reagents. This hinders the acquisition of scientific knowledge regarding the virus-inactivating and viral RNA-stabilizing activities of these buffers, which are directly connected to the safety and reliability of molecular diagnosis. In this study, we evaluated the performance of Prep Buffer A, a commercially available virus lysis/transport buffer for viral RNA provided by Precision System Science Co., Ltd. (Matsudo, Japan). The buffer contains guanidine hydrochloride (GuHCl), a chaotropic agent known to inactivate some enveloped viruses [28,29], and hexadecyltrimethylammonium chloride (Hexa-DTMC), a surfactant with inactivating activity against enveloped viruses [30]. More precisely, this study evaluated the SARS-CoV-2–inactivating activity of Prep Buffer A and its individual components and their ability to stabilize viral RNA copy numbers in SARS-CoV-2-spiked saliva at different temperatures. Although it is recommended that clinical samples for viral diagnosis should be transported and stored at 2–8 °C before testing, it is not always feasible to adhere to these conditions. Temperature control inadequacy has been witnessed, especially during the high infection waves where storage facilities were totally utilized, necessitating sample storage at room temperatures until they were tested. Moreover, in Japan and possibly in many other countries, saliva samples collected by individuals at home may have been transported to the diagnostic laboratories at ambient temperatures, which could be at ≥35 °C in the summer season. Therefore, this study included a storage condition at 35 °C. Additionally, we explored the direct effects of the buffer and its individual components on naked (non-capsulated) SARS-CoV-2 RNA.

## 2. Materials and Methods

### 2.1. Test Solutions

Prep Buffer A, which contains 6.0 M GuHCl and 1.5% (*w*/*v*) Hexa-DTMC, was provided by Precision System Science Co., Ltd. Meanwhile, 6.0 M GuHCl (Tokyo Chemical Industry Co., Ltd., Tokyo, Japan) and 1.5% (*w*/*v*) Hexa-DTMC (Tokyo Chemical Industry Co., Ltd.) were prepared by dissolving the chemicals in ultra-pure water (UPW). All three test solutions (Prep Buffer A, GuHCl, and Hexa-DTMC) and UPW (negative control) were mixed with phosphate-buffered saline (PBS) at a ratio of 5:8, giving Prep Buffer A/PBS, GuHCl/PBS, Hexa-DTMC/PBS, and UPW/PBS.

### 2.2. Viruses and Cells

The 2019-nCoV/Japan/TY/WK-521/2020 strain of SARS-CoV-2 (GISAID ID: EPI_ISL_408667) was kindly provided by the National Institute of Infectious Diseases (Tokyo, Japan). VeroE6/TMPRSS2 cells [31] were obtained from the Japanese Collection of Research Bioresources (Osaka, Japan; cell no. JCRB1819). The passaging of VeroE6/TMPRSS2 cells and cultivation of SARS-CoV-2–inoculated VeroE6/TMPRSS2 cells were performed as previously described [32]. All virus manipulations before inactivation were conducted in a biosafety level 3 facility.

### 2.3. Evaluation of the SARS-CoV-2-Inactivating Activity of Prep Buffer A

SARS-CoV-2 solution containing 1% or 50% fetal bovine serum (FBS) (viral titer: 7.0 log_10_ 50% tissue culture infective dose (TCID_50_)/mL), which simulated virus solutions containing different amounts of organic substances, was mixed with Prep Buffer A/PBS, GuHCl/PBS, or Hexa-DTMC/PBS at a ratio of 7:13 (*n* = 3–6 per group). In the mixtures, the final concentration of GuHCl and percentage of Hexa-DTMC were 1.5 M and 0.384% (*w*/*v*), respectively. As a negative control, the SARS-CoV-2 solution was mixed with UPW/PBS. These mixtures were incubated at 25 °C for 5 min and then loaded into Pierce Detergent Removal Spin Columns (Thermo Fisher Scientific, Rockford, IL, USA) to remove cytotoxic compounds in test solutions. After spin column treatment, the mixtures were inoculated into VeroE6/TMPRSS2 cells followed by 10-fold serial dilution in a cell culture medium. After 1 h of incubation, the cell culture medium was replaced with a virus-free fresh medium, followed by 3 days of incubation. The cytopathic effect on the cells was observed, and the viral titer (log_10_ TCID_50_/mL) was calculated using the Behrens–Kärber method [33]. The virus-inactivating activities of Prep Buffer A, GuHCl, and Hexa-DTMC were evaluated by comparing the viral titers in these test solution groups with that in the UPW group.

### 2.4. RNA Extraction and SARS-CoV-2 Gene Detection

SARS-CoV-2 diluted with PBS was spiked into saliva (Normal Saliva, Pooled Human Donors, Lee Biosolutions Inc., Maryland Heights, MO, USA). SARS-CoV-2-spiked saliva containing various viral titers was mixed with Prep Buffer A at a ratio of 7:13 and immediately subjected to RNA extraction using a QIAamp Viral RNA Kit (QIAGEN N.V., Venlo, The Netherlands) according to the manufacturer’s instructions. The extracted RNA was subjected to real-time one-step RT-PCR in a LightCycler^®^ 96 System (Roche Diagnostics, Basel, Switzerland) using a QuantiTect Probe RT-PCR Kit (QIAGEN). The primers and probe used were as follows: NIID_2019-nCOV_N_F2, 5′-AAATTTTGGGGACCAGGAAC-3′; NIID_2019-nCOV_N_R2ver3, 5′-TGGCACCTGTGTAGGTCAAC-3′; and NIID_2019-nCOV_N_P2, FAM-ATGTCGCGCATTGGCATGGA-TAMRA [34]. The RT-PCR conditions were as follows: reverse transcription (one cycle) at 50 °C for 30 min; denaturation (one cycle) at 95 °C for 15 min; and real-time PCR (45 cycles) at 95 °C for 15 s and 60 °C for 1 min. These procedures were conducted according to the SARS-CoV-2 gene detection manual (ver. 1.1) of the National Institute of Infectious Diseases. The limit of detection (LOD) for the viral titer (viral titer LOD) of SARS-CoV-2-spiked saliva was determined as the viral titer (TCID_50_/mL) of SARS-CoV-2-spiked saliva samples for which the minimum viral gene level could be detected with 95% probability. The viral titer LOD was calculated using a simple logistic regression model plotted in GraphPad Prism ver. 9.3.1 (GraphPad Software Inc., San Diego, CA, USA) and determined to be 2.16 (0.335 log_10_) TCID_50_/mL (Appendix A).

### 2.5. Evaluation of Viral Genome Stability in Test Solutions

Saliva spiked with 1 or 3 log_10_ TCID_50_/mL SARS-CoV-2 or PBS-spiked saliva, which was a virus-free control, was mixed with Prep Buffer A/PBS, GuHCl/PBS, or Hexa-DTMC/PBS at a ratio of 7:13 (*n* = 4 per group). In the mixtures, the final concentration of GuHCl and percentage of Hexa-DTMC were 1.5 M and 0.384% (*w*/*v*), respectively. As a negative control, saliva samples were mixed with UPW/PBS. Using a QIAamp Viral RNA Kit, RNA was extracted from these mixtures immediately (day 0) or after incubation for 1 or 7 days. The incubation conditions were as follows: 35 °C for 7 days, 35 °C for the first 3 days followed by 4 °C for 4 days, or 4 °C for 7 days. These RNA extracts were subjected to real-time one-step RT-PCR as described previously. Ten-fold diluted Takara SARS-CoV-2 Positive Control (RNA) (Takara Bio Inc., Kusatsu, Japan) with 1–6 log_10_ copies/μL and a negative control were also loaded in every real-time one-step RT-PCR. A standard curve was plotted and used to determine the viral gene copy numbers in 1 μL of RNA extract derived from each saliva sample. The LOD for the viral gene copy number (viral gene copy LOD) in this real-time one-step RT-PCR was determined by analyzing 2-fold serial dilutions of Takara SARS-CoV-2 Positive Control from 100 to 1.625 copies/μL. The viral gene copy LOD was determined as the viral gene copy number per microliter of the positive control sample in which the minimum number of viral gene copies could be detected with 95% probability. This viral gene copy LOD was calculated using a simple logistic regression model as previously described and determined as 69.984 (1.845 log_10_) copies/μL (Appendix A).

In other experiments, SARS-CoV-2-spiked saliva with similar viral titers as the aforementioned samples was mixed with Prep Buffer A/PBS at a ratio of 7:13 (*n* = 4 per group). Immediately after mixing (day 0), or after 3 or 7 days of incubation at 4 °C or 35 °C, nucleic acid extraction and real-time RT-PCR targeting SARS-CoV-2 Orf 1ab and N genes were consecutively performed with geneLEAD VIII (Precision System Science Co., Ltd.), an automatic sample-to-result precision instrument, using MagDEA Dx SV (Precision System Science Co., Ltd.) and LeaDEA VIASURE SARS-CoV-2 PCR kit (CerTest Biotech, S.L. Zaragoza, Spain), respectively.

### 2.6. Direct Effect of Prep Buffer A on SARS-CoV-2 RNA

SARS-CoV-2 RNA extracted from a SARS-CoV-2 solution with a viral titer of 7.0 log_10_ TCID_50_/mL was mixed with Prep Buffer A/PBS, GuHCl/PBS, or Hexa-DTMC/PBS at a ratio of 7:13 in the presence or absence of RNase A (Roche Diagnostics, Mannheim, Germany) (*n* = 8 per group). In the mixture, the final concentration of GuHCl and percentage of Hexa-DTMC were 1.5 M and 0.384% (*w*/*v*), respectively. As a negative control, SARS-CoV-2 RNA was mixed with UPW/PBS. In the RNase A-supplemented mixture, the final concentration of RNase A was 100 pg/mL. Then, real-time one-step RT-PCR was performed immediately (day 0) or after 3 or 7 days of incubation at 4 °C or 35 °C to analyze the changes in the viral genome copy numbers in the mixtures.

### 2.7. Statistical Analysis

To evaluate the virucidal activity of each test solution, the unpaired *t*-test was performed for comparisons between the UPW group and each test solution group, and the Kruskal–Wallis test with Dunn’s multiple comparison test was performed for comparisons among three or more groups. To evaluate viral genome stability in test solutions, the unpaired *t*-test was performed for comparisons between two groups, and the Kruskal–Wallis test with Dunn’s multiple comparison test was performed for comparisons among three or more groups.

## 3. Results

### 3.1. Virucidal Activity of Prep Buffer A against SARS-CoV-2

After 5 min of incubation at 25 °C, Prep Buffer A demonstrated virucidal activity against SARS-CoV-2 solution containing 1% FBS, reducing viral titers by ≥4.0 log_10_ TCID_50_/mL compared to UPW (≥99.990% viral inactivation). Prep Buffer A also exhibited virucidal activity against SARS-CoV-2 solution with high organic substance loading (50% FBS). In this case, Prep Buffer A reduced viral titers by ≥3.333 log_10_ TCID_50_/mL (≥99.954% viral inactivation). Because the efficiency of cytotoxic compounds removal by the Pierce Detergent Removal Spin Columns was affected by high organic loading, the LOD of viral titers in the virucidal test (LOD in virucidal test) differed between 1% and 50% FBS (Figure 1A). Additionally, the virucidal activities of the individual Prep Buffer A components (GuHCl and Hexa-DTMC) were evaluated against a SARS-CoV-2 solution containing 1% FBS. GuHCl treatment reduced viral titers by 0.67 log_10_ TCID_50_/mL (78.620% viral inactivation), which was not significantly different from the findings in the UPW group. Conversely, Hexa-DTMC treatment reduced viral titers by ≥4.33 log_10_ TCID_50_/mL (99.995% viral inactivation; the viral titer was below the detection limit) which was statistically significant compared to the UPW group. Prep Buffer A treatment, which is a combination GuHCl and Hexa-DTMC, reduced viral titers by 3.58 log_10_ TCID_50_/mL (99.974% viral inactivation), which was significantly better than the effect of UPW. Virucidal activity did not differ between the Hexa-DTMC and Prep Buffer A groups or between the GuHCl and Prep Buffer A groups, but the difference was significant between the GuHCl and Hexa-DTMC groups (Figure 1B). This result suggests that Hexa-DTMC is the potent compound responsible for SARS-CoV-2 inactivation by Prep Buffer A, and the virucidal activity of Hexa-DTMC tends to be reduced in the presence of GuHCl.

### 3.2. Stability of Viral RNA in SARS-CoV-2-Spiked Saliva Treated with Prep Buffer A

SARS-CoV-2-spiked saliva samples with a high (3 log_10_ TCID_50_/mL; 463 × viral titer LOD) or low viral titer (1 log_10_ TCID_50_/mL; 4.63 × viral titer LOD) were treated with UPW or Prep Buffer A and incubated at 35 °C. The chronological change of viral gene copy numbers in the mixtures was evaluated. The Prep Buffer A (virus-free) group was used as the negative control. After 7 days of incubation at 35 °C, the reduction in viral gene copy numbers was greater in the Prep Buffer A group than in the UPW group irrespective of the viral titer spiked in the saliva. When saliva samples with a high viral titer (3 log_10_ TCID_50_/mL) were tested, the percentage reductions in viral gene copy numbers at day 7 versus day 0 were 70.48% and 98.67% in the UPW and Prep Buffer A groups, respectively. When the saliva samples with a low viral titer (1 log_10_ TCID_50_/mL) were tested, the percentage reductions in these groups were 69.66% and ≥92.02%, respectively (Figure 2).

Viral RNA stability was evaluated when the SARS-CoV-2-spiked saliva samples treated with Prep Buffer A were stored at 35 °C for 7 days, 35 °C for the first 3 days followed by 4 °C for 4 days (35/4°C), or 4°C for 7 days. When saliva samples with a high viral titer (3 log_10_ TCID_50_/mL) were tested, the reductions in viral gene copy number after 7 days were 28.93%, 76.18%, and 96.16% in the 4 °C, 35/4 °C, and 35 °C groups, respectively. Similarly, when saliva samples with a low viral titer (1 log_10_ TCID_50_/mL) were tested, the percentage reductions in these groups were 24.63%, ≥75.38%, and ≥83.79%, respectively. In both the high and low viral titer groups, viral gene copy numbers at day 7 significantly differed between the 4 °C and 35 °C conditions. Conversely, the difference was not significant between the 4 °C and 35/4 °C groups or between the 35/4 °C and 35 °C groups (Figure 3).

The RNA stability of SARS-CoV-2-spiked saliva samples with a high (3 log_10_ TCID_50_/mL) or low viral titer (1 log_10_ TCID_50_/mL) treated with Prep Buffer A was then evaluated using the geneLEAD VIII platform. The samples treated with Prep Buffer A were incubated at either 35 °C or 4 °C over a 7-day period. The storage temperature had a statistically significant role in stabilizing viral RNA in SARS-CoV-2-spiked samples over the 7-day period. Incubation at 35 °C resulted in a greater increase in the Ct value than that at 4 °C in real-time RT-PCR targeting both the Orf 1ab and N genes (Figure 4A,B).

The effect of the difference of saliva on viral RNA stability in Prep Buffer A was evaluated. Three different lots of SARS-CoV-2-spiked saliva (lots 1–3) with a high viral titer (3 log_10_ TCID_50_/mL) were treated with Prep Buffer A and stored at 35 °C for 7 days. The reductions in viral gene copy numbers were 98.57%, 99.16%, and 98.56% in lots 1, 2, and 3, respectively. Although the viral gene copy number in lot 2 was statistically different from those in lots 1 and 3 at day 7, the reduction patterns were generally similar (Appendix A). Thus, the low stability of the SARS-CoV-2 genome in Prep Buffer A at 35 °C was independent of the saliva lot used.

### 3.3. Stability of Viral RNA in SARS-CoV-2-Spiked Saliva Treated with GuHCl and Hexa-DTMC

The viral RNA stabilities in SARS-CoV-2-spiked saliva samples with a high (3 log_10_ TCID_50_/mL) or low viral titer (1 log_10_ TCID_50_/mL) treated with UPW, GuHCl, Hexa-DTMC, and Prep Buffer A were compared. When saliva samples with a high viral titer were tested, GuHCl and Hexa-DTMC treatment for 7 days at 35 °C reduced viral gene copy numbers by 98.24% and 97.59%, respectively, which was nearly similar to the reduction of 98.80% in the Prep Buffer A group, whereas UPW treatment reduced viral gene copy numbers by 71.65%. The multiple comparison test revealed a significant difference at day 7 between the UPW group and the GuHCl, Hexa-DTMC, and Prep Buffer A groups as well as between the Hexa-DTMC and Prep Buffer A groups. When saliva samples with a low viral titer (1 log_10_ TCID_50_/mL) were tested, the viral gene copy numbers in the GuHCl, Hexa-DTMC, and Prep Buffer A groups were below the viral copy LOD at day 7 (≥70.0% reduction), whereas the percentage reduction was 58.0% in the UPW group. There were no significant differences among these groups (Figure 5).

### 3.4. Direct Effects of Prep Buffer A and Its Individual Components on Naked SARS-CoV-2 RNA

The direct effects of GuHCl, Hexa-DTMC, and Prep Buffer A on naked SARS-CoV-2 RNA and their inhibitory effects on RNase were evaluated. Viral RNA extracted from a SARS-CoV-2 solution was treated with UPW, GuHCl, Hexa-DTMC, or Prep Buffer A and stored at 35 °C for 7 days in the presence or absence of RNase. The percentage reductions in viral gene copy numbers after 7 days were ≥99.98% and 96.90% in the UPW/RNase (+) and UPW/RNase (−) groups, respectively (Appendix A). Conversely, the percentage reductions after 7 days were 71.91% and 67.47% in the GuHCl/RNase (+) and GuHCl/RNase (−) groups, respectively, 99.50% and 98.95% in the Hexa-DTMC/RNase (+) and Hexa-DTMC/RNase (−) groups, respectively, and 94.79% and 96.07% in the Prep Buffer A/RNase (+) and Prep Buffer A/RNase (−) groups, respectively. The differences were significant in the GuHCl/RNase (+) and Hexa-DTMC/RNase (+) groups, the GuHCl/RNase (+) and Hexa-DTMC/RNase (−) groups, the GuHCl/RNase (−) and Hexa-DTMC/RNase (+) groups, and the GuHCl/RNase (−) and Hexa-DTMC/RNase (−) groups (Figure 6). The results suggest that the reduction in viral gene copy number was smaller in the GuHCl group than in the Hexa-DTMC group, and the presence of RNase did not affect the reduction by GuHCl, Hexa-DMTC, and Prep Buffer A.

## 4. Discussion

To ensure biosafety during SARS-CoV-2 diagnostics, guanidine-based virus lysis/transport buffers, including those containing guanidine thiocyanate (GTC) or GuHCl, have been widely explored for inactivating SARS-CoV-2 [25,27,35,36]. Previously, reagents containing these salts were reported to effectively inactivate multiple virus species including the Middle East respiratory syndrome coronavirus [37], Ebola virus [38], and other enveloped viruses [39]. In this study, we evaluated the performance of Prep Buffer A, a virus lysis/transport buffer for viral RNA containing GuHCl and a surfactant. The buffer inactivated SARS-CoV-2 within 5 min in the presence of 1% FBS (≥4.0 log_10_ TCID_50_/mL reduction in viral titer; ≥99.990% viral inactivation), similar to the findings for other reagents containing both GuHCl and surfactants [25,27,36]. The virucidal activities of many disinfectants are reduced in the presence of organic substances [40]. The reduction in the titers of SARS-CoV-2 treated with Prep Buffer A was slightly smaller in the presence of 50% FBS than that for SARS-CoV-2 in the presence of 1% FBS (Figure 1A). Although the presence of high organic substance content appears to slightly reduce the virucidal activity of Prep Buffer A, these results indicate that Prep Buffer A enhances biosafety during the collection, packaging, transportation, and handling of saliva samples for diagnosing SARS-CoV-2 infection by RT-PCR. In several guanidine-based lysis buffers, GuHCl or GTC is routinely used at a working concentration of 4–6 M (before mixing with virus-containing samples) [28,36,41,42]. The final concentration for virus inactivation depends on prior dilution with other reagents, if any, as well as the virus-containing sample-to-buffer ratio. In previous studies evaluating the SARS-CoV-2-inactivating activity of guanidine-based lysis buffers, the final concentration of guanidine salt ranged from 0.37 M [36] to 3.64 M [27]. The buffer containing GuHCl, Triton X-100, Tris-HCl, and EDTA-Na reduced the SARS-CoV-2 titer by >3.7 log_10_ TCID_50_/mL in 2 min; the final concentrations of GuHCl and Triton X-100 were 0.37 M and 0.09%, respectively [36]. Some GTC-based [27,41,43,44] and GuHCl-based lysis buffers [27,36] have demonstrated SARS-CoV-2-inactivating activity to various degrees. However, these buffers contain other potentially virucidal chemicals, including detergents and solvents such as alcohols. To the best of our knowledge, no previous report has evaluated the virucidal activity of GuHCl alone against SARS-CoV-2. In this study, when the inactivating roles of the individual components of Prep Buffer A were evaluated, GuHCl alone, at a final concentration of 1.5 M in the infectious samples, did not exhibit significant virucidal activity against SARS-CoV-2. Furthermore, the virucidal activity of Hexa-DTMC was slightly reduced in the presence of GuHCl. Although we evaluated only one concentration of GuHCl, its activity may be improved by increasing its final concentration. Thom et al. [44] showed that a GTC-based buffer with a higher GTC concentration exerted more potent virucidal activity against SARS-CoV-2 than a buffer with a lower GTC concentration. However, as mentioned above, components other than GTC also differed between these two buffers. When considered together, these findings suggest that the constituents of guanidine-based lysis buffers and their concentrations influence the SARS-CoV-2-inactivating activity of the buffers.

The immediate collection and diagnosis of samples from patients with suspected COVID-19 are paramount for clinical management and infection control measures [45]. However, in many low- and middle-income countries, challenges including, but not limited to, a lack of equipment, reagents, and healthcare workers trained for COVID-19 diagnostics [46] may lead to an overflow of samples, necessitating their preservation until testing. In this study, we evaluated the effect of Prep Buffer A on the stability of SARS-CoV-2 RNA by comparing viral genome copy numbers over 7 days and varying temperature conditions to reflect practical sample storage situations. We examined stability by storing samples at 35 °C to reflect temperatures during specimen transportation in dry tropical regions or extreme summer temperatures in temperate zones, even in Japan. The reduction in viral genome copy numbers was significantly different between the Prep Buffer A and UPW groups (Figure 2). This instability of SARS-CoV-2 RNA in Prep Buffer A at 35 °C coincided with that in other guanidine-based buffers such as eNAT at 35 °C [43]. Furthermore, GuHCl and Hexa-DTMC alone also reduced viral gene copy numbers at 35 °C (Figure 5). Conversely, SARS-CoV-2 RNA was relatively stable in non-treated saliva for prolonged periods at 30 °C [16]. Therefore, if the temperature cannot be controlled during sample transportation, the transport of non-treated saliva might be more appropriate in situations in which alternative biosafety measures are available. When the samples were stored under the 35/4 °C condition to mimic a situation in which samples are collected in Prep Buffer A and transported to the laboratory without a cold chain within 3 days but immediately stored at 4 °C after reaching the laboratory, the viral genome reduction was considerably improved compared to the findings at 35 °C (Figure 3). However, the best condition for sample transportation and storage with Prep Buffer A was assumed to be in a cold chain (4 °C), in which viral genome copy numbers were reduced by approximately 25% at day 7. This high viral RNA stability in Prep Buffer A at 4 °C was also observed in automatic sample-to-result analysis using geneLEAD VIII (Figure 4). This geneLEAD VIII system has proven effective and reliable in the implementation of SARS-CoV-2 laboratory diagnostic tests in the sample-to-result workflow [47]. The results suggest that the collecting and storing of saliva samples in Prep Buffer A at 4 °C offers great advantages.

As saliva contains RNases, which have been reported to affect the detection of the SARS-CoV-2 genome [48], the impact of different saliva lots on the degradation of viral RNA in Prep Buffer A at 35 °C was evaluated. The results were similar among the lots. Meanwhile, the presence or absence of RNase did not affect the activities of GuHCl, Hexa-DTMC, and Prep Buffer A against naked viral RNA. These results indicated that both GuHCl and Hexa-DTMC have RNase-inhibitory activity. RNA stability was better in the GuHCl/RNase (+) group than in the Hexa-DTMC/RNase (+), Prep Buffer A/RNase (+), and UPW/RNase (+) groups. On the contrary, the chronological reduction in copy numbers was larger in the Hexa-DTMC/RNase (−) group than in the GuHCl/RNase (−) and UPW/RNase (−) groups (Figure 6 and Appendix A). These results suggest that Hexa-DTMC directly destroyed naked SARS-CoV-2 RNA at 35 °C. However, the results shown in Figure 5 and Figure 6 illustrate that the reduction in viral genome copy numbers in GuHCl-treated SARS-CoV-2-spiked saliva at 35 °C was not attributable to the direct destruction of viral RNA. One possible hypothesis, although no confirmatory data were obtained in this study, is that viral RNA leaked from GuHCl-treated vulnerable virus particles, after which it was degraded in a GuHCl- and RNase-independent manner under unstable naked (non-capsulated) conditions.

This study had several limitations. First, we did not evaluate the performance of Prep Buffer A on saliva from patients with COVID-19. Instead, we used artificial virus-spiked saliva samples with high and low viral titers (463 and 4.63 × viral titer LOD, respectively) in this study, which mimics the broad range of viral RNA levels in actual samples derived from patients with COVID-19 [49]. Second, because the buffer is only intended for collecting saliva samples, we did not evaluate its performance on other respiratory samples used for SARS-CoV-2 diagnosis.

## 5. Conclusions

We found that 0.384% (w/v) Hexa-DTMC, the surfactant in Prep Buffer A, but not 1.5 M GuHCl, effectively inactivated SARS-CoV-2, suggesting that Hexa-DTMC was essential in Prep Buffer A to potently inactivating SARS-CoV-2. The preservation of SARS-CoV-2-spiked saliva samples in Prep Buffer A at 4 °C offered the best viral RNA stability for 7 days, as viral levels were reduced by only approximately 25%. Although both GuHCl and Hexa-DTMC inhibited RNase activity, the chronological reduction in viral gene copy numbers was induced in both GuHCl- and Hexa-DTMC-treated SARS-CoV-2-spiked saliva at 35 °C. Our findings suggest that samples collected in Prep Buffer A should be stored at 4 °C when real-time RT-PCR will not be performed for several days. This study provides useful and detailed knowledge on SARS-CoV-2 inactivation and viral RNA stability in a guanidine-based buffer containing surfactant, which are widely used as a virus lysis/transport buffer.

## Figures and Tables

**Figure 1 viruses-15-00509-f001:**
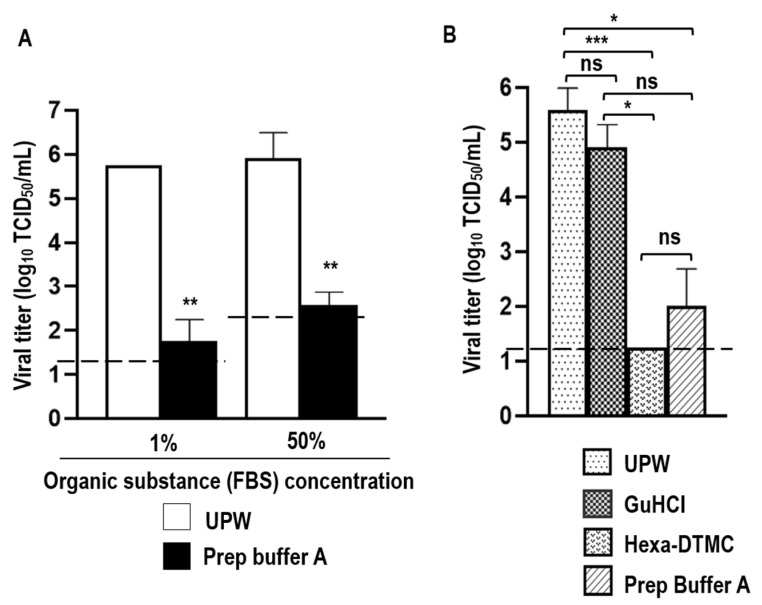
SARS-CoV-2-inactivating activity of Prep Buffer A and/or its individual components. (**A**) Virucidal effects of Prep Buffer A against SARS-CoV-2 solution containing 1% or 50% FBS after 5 min of incubation. The unpaired *t*-test was performed to analyze differences between the UPW and Prep Buffer A groups (*n* = 3 per group; ** *p* < 0.01). (**B**) Virucidal effects of GuHCl, Hexa-DTMC, and Prep Buffer A against SARS-CoV-2 solution containing 1% FBS after 5 min of incubation. The Kruskal–Wallis test with Dunn’s multiple comparison test was performed to analyze the significance of differences among all tested groups (*n* = 6 per group; * *p* < 0.05; *** *p* < 0.001; ns: not significant). The black dashed lines indicate the LOD in virucidal tests (1.25 log_10_ TCID_50_/mL in the 1% FBS condition; 2.25 log_10_ TCID_50_/mL in the 50% FBS condition).

**Figure 2 viruses-15-00509-f002:**
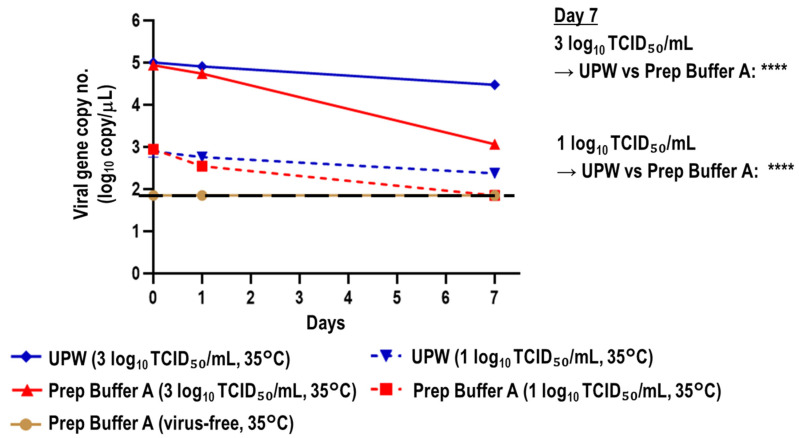
Stability of viral RNA in SARS-CoV-2-spiked saliva treated with UPW or Prep Buffer A at 35 °C. SARS-CoV-2-spiked saliva with high or low viral titers (3 or 1 log_10_ TCID_50_/mL, respectively) was treated with UPW or Prep Buffer A and stored at 35 °C for 7 days. Virus-free saliva treated with Prep Buffer A was used as a control. The unpaired *t*-test was performed to analyze the significance of differences in viral gene copy numbers at day 7 between the UPW (3 log_10_ TCID_50_/mL) and Prep Buffer A groups (3 log_10_ TCID_50_/mL) and between the UPW (1 log_10_ TCID_50_/mL) and Prep Buffer A groups (1 log_10_ TCID_50_/mL; *n* = 4 per group; **** *p* < 0.0001). The black dashed line indicates the viral gene copy LOD (1.845 log_10_ copies/μL).

**Figure 3 viruses-15-00509-f003:**
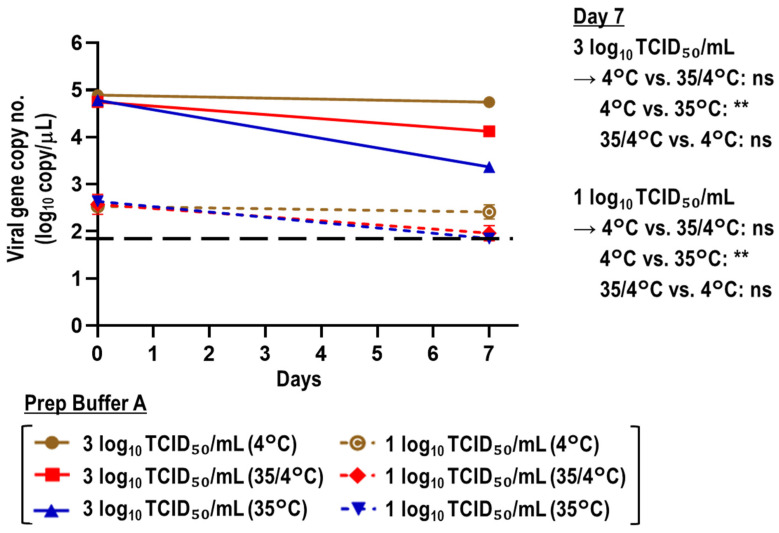
Stability of viral RNA in SARS-CoV-2-spiked saliva treated with Prep Buffer A at different temperatures. SARS-CoV-2-spiked saliva with high or low viral titers (3 or 1 log_10_ TCID_50_/mL, respectively) was treated with Prep Buffer A and stored at 35 °C, 35/4 °C, or 4 °C for 7 days. The Kruskal–Wallis test with Dunn’s multiple comparison test was performed to analyze the significance of differences in viral gene copy numbers at day 7 among the different temperature conditions (*n* = 4 per group; ** *p* < 0.01; ns: not significant). The black dashed line indicates the viral gene copy LOD (1.845 log_10_ copies/μL).

**Figure 4 viruses-15-00509-f004:**
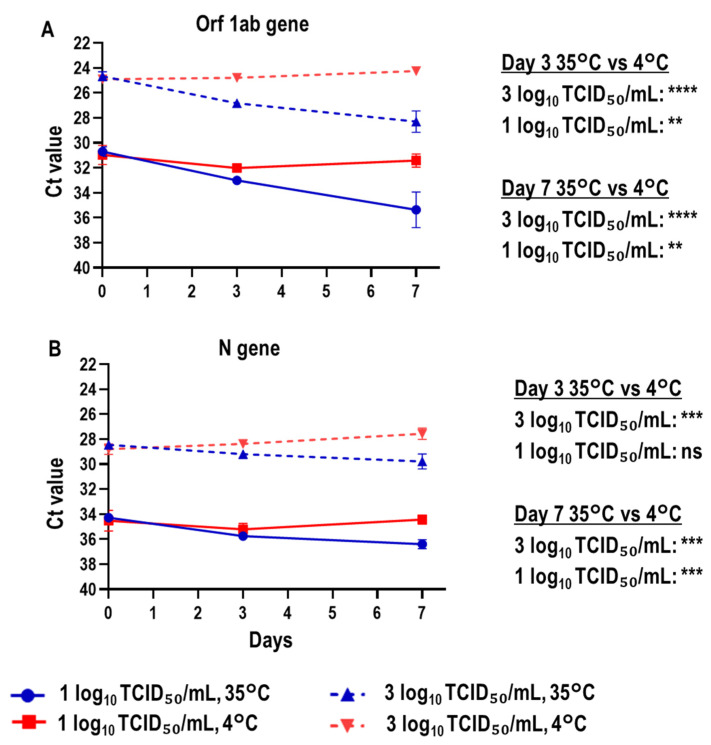
Stability of viral RNA in SARS-CoV-2-spiked saliva treated with Prep Buffer A at 35 °C and 4 °C as analyzed using geneLEAD VIII. (**A**,**B**) SARS-CoV-2-spiked saliva with high or low viral titers (3 or 1 log_10_ TCID_50_/mL, respectively) was treated with Prep Buffer A and stored at 35 °C or 4 °C for 7 days. Automatic nucleic acid extraction and real-time RT-PCR targeting the SARS-CoV-2 Orf 1ab (**A**) and N (**B**) genes were consecutively performed using the geneLEAD VIII platform. The unpaired *t*-test was performed to analyze the significance of differences between the 35 °C and 4 °C groups for low and high viral titers at days 3 and 7 (*n* = 4 per group; ** *p* < 0.01; *** *p* < 0.001; **** *p* < 0.0001; ns: not significant).

**Figure 5 viruses-15-00509-f005:**
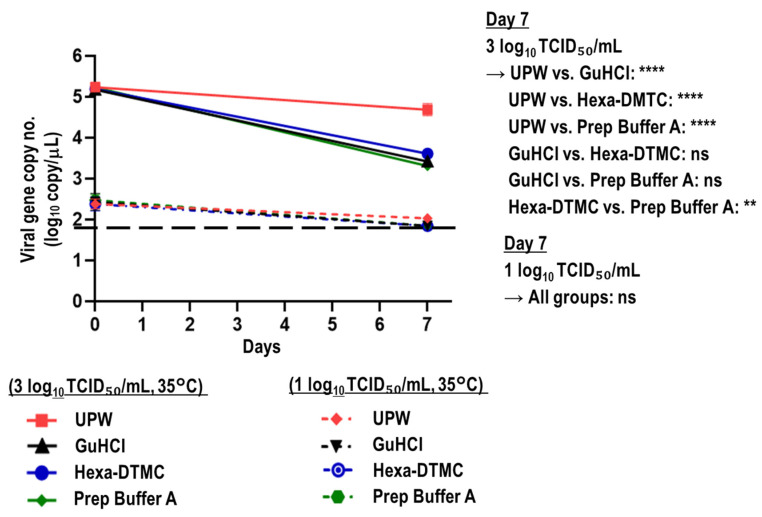
Stability of viral RNA in SARS-CoV-2-spiked saliva treated with GuHCl and Hexa-DTMC. SARS-CoV-2-spiked saliva with high or low viral titers (3 or 1 log_10_ TCID_50_/mL, respectively) was treated with UPW, GuHCl, Hexa-DMTC, or Prep Buffer A and stored at 35 °C for 7 days. The Kruskal–Wallis test with Dunn’s multiple comparison test was performed to analyze the significance of differences in viral gene copy numbers at day 7 among the different test solution groups (*n* = 4 per group; ** *p* < 0.01; **** *p* < 0.0001; ns: not significant). The black dashed line indicates the viral gene copy LOD (1.845 log_10_ copies/μL).

**Figure 6 viruses-15-00509-f006:**
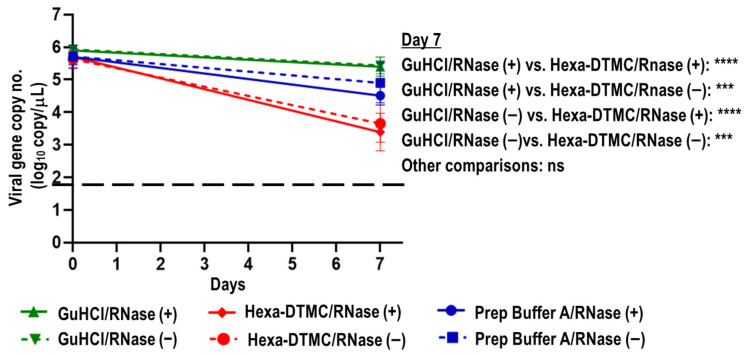
Direct impact of Prep Buffer A and its individual components on SARS-CoV-2 RNA. Viral RNA extracted from a SARS-CoV-2 solution was treated with GuHCl, Hexa-DMTC, or Prep Buffer A and stored at 35 °C for 7 days in the presence or absence of RNase. The Kruskal–Wallis test with Dunn’s multiple comparison test was performed to analyze the significance of differences in viral gene copy numbers at day 7 among the different test solution groups (*n* = 8 per group; *** *p* < 0.001; **** *p* < 0.0001; ns: not significant). The black dash line indicates the viral gene copy LOD (1.845 log_10_ copies/μL).

## Data Availability

The original contributions presented in the study are included in the article and Appendix A. Further inquiries can be directed to the corresponding author.

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
