# Peer review of "Efficacy Validation of SARS-CoV-2-Inactivation and Viral Genome Stability in Saliva by a Guanidine Hydrochloride and Surfactant-Based Virus Lysis/Transport Buffer"

_viruses, 2023, doi:10.3390/v15020509_

Round 1

Reviewer 1 Report

This paper reports an investigation of a proprietary buffer Prep Buffer A into the inactivation of SARS-COV-2 in diagnostic laboratory settings, principally from saliva samples. In more detail the study investigates the inactivation efficacy of two ingredients of this buffer; the detergent Hexa-DTMC and the guanidine salt GuHCL.

The paper is well written, clear to follow and presents some interesting findings that will add to the already extensive body of work on the inactivation of SARS-COV-2 in diagnostic laboratory settings. However, whilst the paper present findings that are consistent with the published body of work (detergent based components are an important factor in SARS-COV-2 inactivation) the finding that GuHCL is not an important inactivant of SARS-COV-2 contradicts findings from many other studies. This is not adequatley addressed in the manuscript.

Specific comments:

Page 1:, line 29. The line that says ‘The role of GuHCl in virus-lysis/transport buffers for SARS-CoV-2 was insignificant’ is a sweeping statement that is both highly significant in itself but whose potential significance is not addressed in the manuscript or backed up with evidential reference. In addition to the papers cited there are several other papers (i.e. Thom et al., 2021.

doi.org/10.3389/fcimb.2021.716436; Banik et al., 2021. doi.org/10.1371/journal.pone.0252687) where at least partial virucidal activity (against SARS-COV-2) of guanidinium based chaotrophic salts in similar buffers is demonstrated. There is no discussion in the manuscript on this contradiction or the significance of it. For example is the 6M concentration representative chosen in this study representative of all guanidine based buffers as the statement in the abstract seems to suggest? I think that this statement needs modification, appropriate discussion of both its merits and demerits, along with reference against other published studies. This statement is too significant and sweeping for it to be included as it is.  

Author Response

We are greatly indebted to the reviewers for their thorough review of our manuscript and their insightful comments and suggestions. Subsequently, we have attentively revised the manuscript and highlighted all the changes in red. In addition, our point-to-point responses to the reviewers’ comments are provided below:

Reviewer 1

Comment 1: This paper reports an investigation of a proprietary buffer Prep Buffer A into the inactivation of SARS-COV-2 in diagnostic laboratory settings, principally from saliva samples. In more detail the study investigates the inactivation efficacy of two ingredients of this buffer; the detergent Hexa-DTMC and the guanidine salt GuHCL.

The paper is well written, clear to follow and presents some interesting findings that will add to the already extensive body of work on the inactivation of SARS-COV-2 in diagnostic laboratory settings.

Response 1: We thank the reviewer for the positive feedback.

Comment 2: However, whilst the paper present findings that are consistent with the published body of work (detergent based components are an important factor in SARS-COV-2 inactivation) the finding that GuHCl is not an important inactivant of SARS-COV-2 contradicts findings from many other studies. This is not adequately addressed in the manuscript.

Response 2: This comment has been addressed under the response for specific comments.

Comment 3: Page 1: line 29. The line that says ‘The role of GuHCl in virus-lysis/transport buffers for SARS-CoV-2 was insignificant’ is a sweeping statement that is both highly significant in itself but whose potential significance is not addressed in the manuscript or backed up with evidential reference. In addition to the papers cited there are several other papers (i.e. Thom et al., 2021. doi.org/10.3389/fcimb.2021.716436; Banik et al., 2021. doi.org/10.1371/journal.pone.0252687) where at least partial virucidal activity (against SARS-COV-2) of guanidinium based chaotrophic salts in similar buffers is demonstrated. There is no discussion in the manuscript on this contradiction or the significance of it. For example, is the 6M concentration representative chosen in this study representative of all guanidine-based buffers as the statement in the abstract seems to suggest? I think that this statement needs modification, appropriate discussion of both its merits and demerits, along with reference against other published studies. This statement is too significant and sweeping for it to be included as it is.

Response 3: We appreciate the reviewer’s insightful comment. We have modified the statement in the abstract, lines 24–26, as follows: “Hexa-DTMC alone (0.384%), but not 1.5 M GuHCl alone, exhibited considerable virucidal activity, suggesting that it was essential for potently inactivating SARS-CoV-2 using Prep Buffer A.” We have clearly described the precise concentrations of Hexa-DTMC and GuHCl tested in this study to facilitate correct interpretation of the obtained results. In addition, the concentrations of guanidine salts in guanidine-based lysis buffers used in previous studies have been provided in lines 344–352 to aid in the discussion of our results.

As the guanidine-based buffer investigated in this study, Prep Buffer A, showed potent virucidal activity, we do not think our results contradict previous reports wherein other guanidine-based buffers have displayed similar activities. On the contrary, to the best of our knowledge, none of these previous reports evaluated the SARS-CoV-2-inactivating activity of a solution containing guanidine salt alone. Although nowadays, guanidine-based buffers are widely used for the genetic diagnosis of SARS-CoV-2, data on the virucidal activity of guanidine salt alone are lacking. Our study provides basic information on the SARS-CoV-2-inactivating activity of GuHCl. We have discussed these points in lines 349–367 and added more supporting references.

Reviewer 2 Report

Estimated Authors,

I've read with great interest the present study, reporting on the potential SARS-CoV-2 inactivation and viral genome stability in saliva by a guanidine hydrochloridine and surfactant-based virus-lysis/transport buffer. In the present report, through a design study that appears (at least to the present reviewer) high consistent and somehow elegant, Authors have showed that Hexa-DTMC, the surfactant contained in Prep Buffer A, but not GuHCl, effectively inactivated SARS-CoV-2. Authors have also discussed in a straightforward way how they results may be acknowledged as significant in the daily practice. 

From a methodological point of view, I've only the following statistical consideration to be shown to the Authors, and that I would discuss with them before accepting this study.

Across the study, it is quite unclear how many samples were actually collected and examined. Moreover, they employ non-parametric testing for the comparison of the results, acknowledging their non-Gaussian distribution. Still, I'm forced to stress that M-W test does not simply replace student's t test for unpaired data, it is a test with a different meaning (i.e. rank comparison) that is adapted to the settings where the Gaussian pre-requisite is not available. In other words: please provide the actual number of samples performed and reported (particularly in Figure 1), a preventive testing with D'Agostino-Pearson K2 test (or a similar one), and - if Gaussian distribution is not dismissed, please opt for a more reliable Student's t test and similar parametric multiple comparison tests for continuous data.

Author Response

We are greatly indebted to the reviewers for their thorough review of our manuscript and their insightful comments and suggestions. Subsequently, we have attentively revised the manuscript and highlighted all the changes in red. In addition, our point-to-point responses to the reviewers’ comments are provided below:

Reviewer 2

Comment 1: I've read with great interest the present study, reporting on the potential SARS-CoV-2 inactivation and viral genome stability in saliva by a guanidine hydrochloridine and surfactant-based virus-lysis/transport buffer. In the present report, through a design study that appears (at least to the present reviewer) high consistent and somehow elegant, Authors have showed that Hexa-DTMC, the surfactant contained in Prep Buffer A, but not GuHCl, effectively inactivated SARS-CoV-2. Authors have also discussed in a straightforward way how their result may be acknowledged as significant in the daily practice.

Response 1: We thank the reviewer for the positive comment.

Comment 2: From a methodological point of view, I've only the following statistical consideration to be shown to the Authors, and that I would discuss with them before accepting this study.

Across the study, it is quite unclear how many samples were actually collected and examined.

Response 2: The present in vitro study was conducted using SARS-CoV-2 solutions or SARS-CoV-2-spiked saliva prepared in our laboratory. These were not clinical samples. The number of samples tested in each experiment has been added to the respective section in Materials and Methods and is also mentioned in each figure legend. Specifically, in the inactivation study, three (Figure 1A) or six (Figure 1B) independent samples per each test solution group were examined. For the genome stability study, four (Figures 2–5) or eight (Figures 6, S3, S4) independent samples per each test solution group were examined over a 7-day period. This information has been added to the manuscript in lines 110, 148, 167 and 179 and is also described in each figure legend.

Comment 3: Moreover, they employ non-parametric testing for the comparison of the results, acknowledging their non-Gaussian distribution. Still, I'm forced to stress that M-W test does not simply replace student's t test for unpaired data, it is a test with a different meaning (i.e. rank comparison) that is adapted to the settings where the Gaussian pre-requisite is not available. In other words: please provide the actual number of samples performed and reported (particularly in Figure 1), a preventive testing with D'Agostino-Pearson K2 test (or a similar one), and - if Gaussian distribution is not dismissed, please opt for a more reliable Student's t test and similar parametric multiple comparison tests for continuous data.

Response 3: We appreciate the reviewer’s valuable suggestion. We have modified our statistical analysis and used unpaired t-test in Figures 1A, 2, 4, and S4.